# Selection of appropriate reference genes for quantitative real-time reverse transcription PCR in *Betula platyphylla* under salt and osmotic stress conditions

Ziyi Li[1,2☯], Huijun Lu[2☯], Zihang He[2], Chao Wang[2], Yucheng Wang[1], Xiaoyu Ji[1] *

**1** College of Forestry, Shenyang Agricultural University, Shenyang, China, **2** State Key Laboratory of Tree Genetics and Breeding (Northeast Forestry University), Harbin, China

☯ These authors contributed equally to this work.
* jixiaoyu@syau.edu.cn

**Data Availability Statement:** All relevant data are within the manuscript and its Supporting Information files.

## Abstract

Selecting appropriate reference genes is vital to normalize gene expression analysis in birch (*Betula platyphylla*) under different abiotic stress conditions using quantitative real-time reverse transcription PCR (qRT-PCR). In this study, 11 candidate birch reference genes (*ACT*, *TUA*, *TUB*, *TEF*, *18S rRNA*, *EF1α*, *GAPDH*, *UBC*, *YLS8*, *SAND*, and *CDPK*) were selected to evaluate the stability of their expression in different tissues and under different abiotic stress conditions. Three statistical algorithms (GeNorm, NormFinder, and BestKeeper) were used to analyze the stability of the 11 candidate reference genes to identify the most appropriate one. The results indicated that *EF-1α* was the most stable reference gene in different birch tissues, *ACT* was the most stable reference gene for normal conditions, *ACT* and *TEF* were the most stable reference genes for salt stress treatment, *TUB* was the most stable reference gene for osmotic stress treatment, and *ACT* was the most appropriate choice in all samples of birch. In conclusion, the most appropriate reference genes varied among different experimental conditions. However, in this study, *ACT* was the optimum reference gene in all experimental groups, except in the different tissues group. *GAPDH* was the least stable candidate reference gene in all experimental conditions. In addition, three stress-induced genes (*BpGRAS1*, *BpGRAS16*, and *BpGRAS19*) were chosen to verify the stability of the selected reference genes in different tissues and under salt stress. This study laid the foundation for the selection of appropriate reference gene(s) for future gene expression pattern studies in birch.

## Introduction

Quantitative real-time reverse transcription polymerase chain reaction (qRT-PCR) has become a common technique to detect gene expression levels in the field of molecular biology because of its speed, high efficiency, labor-saving, and sensitivity. It has been widely used in

**Funding:** This work was supported by the National Natural Science Foundation of China (No. 31500535), and The Heilongjiang Province Science Foundation (No. QC2018017).

**Competing interests:** The authors have declared that no competing interests exist.

basic biological research [1], food spoilage detection [2], environmental monitoring [3], and diagnosis and treatment of disease [4]. QRT-PCR has strict requirements on the purity and quality of RNA, reverse transcription efficiency, the specificity of the primers, PCR amplification efficiency, and the selection of internal reference genes [5]. Reference genes can have a large influence on the results of relative expression level determination. Thus, selecting ideal reference genes is vital to obtain reliable results. One or more internal reference genes are typically required to normalize qRT-PCR data to obtain more accurate results.

Over the course of an experiment, a reliable internal control should show a minimal change in expression, whereas the expression of a gene of interest may change greatly. Housekeeping genes are often selected because they encode proteins essential to cell viability and therefore are assumed to be expressed similarly in different cell types [6]. Commonly used internal controls include actin (*ACT*), beta-tubulin (*TUB*), elongation factor 1 alpha (*EF1-α*), and glyceraldehyde-3-phosphate dehydrogenase (*GAPDH*) [7–8]. However, increasing numbers of reports have confirmed that the expression of housekeeping genes can vary in different tissues and may be affected by the experimental conditions. For instance, *TUB* is a wildly used reference gene; a study showed that it was unsuitable as a reference gene in different tissues of *Pennisetum glaucum* [9]. The *TUA* (*α-tubulin*) gene was selected as reference gene in *Salicornia europaea* under salt stress [10–11]; whereas it was not considered as an ideal reference gene in *S. europaea* under drought treatment [12]. *ACT* was selected as a reference gene in wheat and *Kosteletzkya virginica* under salt stress, but was not suitable as a reference gene in papaya under many experimental conditions or in *Salicornia europaea* and cucumber under salt stress [12–16]. Therefore, selecting the appropriate reference genes according to different tissues and different experimental conditions is essential.

Studies on the evaluation and validation of reference genes have been conducted in many plant species, especially in herbaceous plants, such as *Arabidopsis thaliana* [17], *Glycine max* [18]. Similar studies have been carried out in certain woody plants, such as *Morus alba* L. [19], *Carex rigescens* [20]. However, the systematic evaluation and validation of appropriate reference genes in birch (*Betula platyphylla*) under different abiotic stress conditions has not been reported.

*Betula platyphylla*, which is a species of deciduous hardwood, is widely distributed in the mid-high mountains of warm temperate regions in the world, including northern China, Russian Far East, Siberia, Mongolia, Northern Korea, and Japan [21]. With the continuous expansion of drought and saline areas worldwide, it is of great significance to cultivate new salt tolerant and drought resistant varieties of birch. Therefore, selection of internal reference genes in birch with stable expression under different abiotic stress conditions plays a key role in studies on stress-related gene expression.

The analysis results of qRT-PCR were affected to varying degrees of influence in reference genes choosing. Some studies show how to choose genes as reference genes to normalized expression data. [22]. Therefore, in this study, eleven candidate reference genes (*ACT*, *TUA*, *TUB*, *TEF* (Translation elongation factor)), *18S rRNA* (18S ribosomal RNA), *EF1α*, *GAPDH*, *UBC* (Ubiquitin-conjugating enzyme E2), *YLS8* (Thioredoxin-like protein YLS8), *SAND* (S-adenosyl methionine decarboxylase), and *CDPK* (CDPK-related kinase) were selected to evaluate their expression stability in different tissues and under abiotic stress conditions in birch. Three different statistical tools (GeNorm, NormFinder, and BestKeeper) were used to analyze the stability of 11 candidate reference genes and identify the most appropriate reference gene. Our results will facilitate of appropriate reference genes selection for the development of gene expression pattern studies in birch.

## Materials and methods

### Plant material and stress treatments

In this experiment, open pollinated northeast white birch (*B. platyphylla*) seeds from the Northeast Forestry University were planted in plastic pots with mixture of turf peat and sand (v/v 3:1), in a growth house where the conditions kept at 70–75% relative humidity, 400 μmol·m$^{-2}$s$^{-1}$ light intensity, a 16 h light/8 h dark photocycle at 25±2˚C. The 8-week-old birch plants grown in soil were watered with a solution of 2 L 200 mM NaCl or 2 L 300 mM mannitol (distilled water served as the control) for 3, 6, 12, 24, and 48 h, respectively. The root, stem, and leaf tissues from 6 birch seedlings were collected after each treatment. Three independent replications were performed. All the samples were placed at −80˚C after freezing in liquid nitrogen for further study.

### Total RNA isolation and cDNA synthesis

The total RNA of all samples was isolated using a Universal Plant Total RNA Extraction Kit (Bioteke, Beijing, China) following the manufacturer's instructions. RNA integrity was checked using 1.5% agarose gel electrophoresis, and the RNA concentration and purity were determined using a NanoDrop 2000 Spectrophotometer (NanoDrop, Thermo Scientific, Waltham, MA, USA). All the RNA samples with A260/A280 ratios = 1.9–2.1 and A260/A230 ratios > 2.0 were used for further study. Then RNA (1 μg) was reversely transcribed into cDNA using a TransScript One-Step gDNA Removal and cDNA Synthesis SuperMIX (TransGen Biotech, Beijing, China) following the manufacturer's instructions. The synthesized cDNA was stored at −20˚C for further study.

### Candidate reference gene selection and primer design

Eleven candidate reference genes that are commonly used in qRT-PCR were chosen for this study *ACT*, *TUA*, *TUB*, *TEF*, *18S rRNA*, *EF1α*, *GAPDH*, *UBC*, *YLS8*, *SAND*, and *CDPK* [22–23]. The GenBank accession numbers for the 11 genes, which were obtained from the birch genome data (unpublished), are listed in Table 1. Specific primers for the reference genes were designed using Primer premier 5.0 (Premier Biosoft International, Palo Alto, CA, USA). The primer sequences for qRT-PCR are listed in Table 1.

### Quantitative real time PCR

A TransStart Top Green qPCR SuperMix kit (TransGen Biotech, Beijing, China) was used for qRT-PCR. All qRT-PCR reactions were carried out on Qtower$^3$G (Analytik, Jena, Germany). The qRT-PCR reaction system included: 10 μL of 2 × *TransStart* Top Green Realtime qPCR SuperMix, 0.4 μL upstream primers, and 0.4 μL downstream primers (10 μM), 2 μL of cDNA templates (100 ng), and 7.2 μL of RNAase-free ddH$_2$O with a volume of 20 μL. The qRT-PCR reaction conditions were as follows: 94˚C for 30 s; 45 cycles at 94˚C for 12 s, 60˚C for 30 s, and 72˚C for 40 s; and 79˚C for 1 s for plate reading. The melting curve was constructed to verify the specificity of each sequence-specific primer. The PCR amplification efficiency (E) was calculated as E = −1 + 10 (−1/slope) [24], where the slope was derived from a standard curve generated by 10-fold serial dilutions of the mixed cDNA for each primer pair. Three independent replications were performed.

### Data analysis

SigmaPlot 10 software (Systat Software, Inc., San Jose, CA, USA) was used to show the cycle threshold (Ct) value distribution of all samples, which can reflect clearly the average expression level of the genes in the samples. Threshold for the Ct values is the machine setting. The average Ct value was calculated using three biological replications. GeNorm [25] and NormFinder

**Table 1. Primer sequences and amplification parameters of 11 candidate reference genes from birch.**

| Gene symbol | Gene name | Accession no. | Primers (5′-3′) | Tm | E | Amplicon length (bp) | Mean Ct | SD | CV (%) |
|---|---|---|---|---|---|---|---|---|---|
| 18S rRNA | 18S ribosomal RNA | MK388236 | F: AACGAACGAGACCTCAGCCT<br>R: ACTCGTTGAATACATCAGTG | 61.18<br>53.09 | 1.97 | 171 | 17.39 | 1.36 | 14.01 |
| GAPDH | Glyceraldehyde-3-phosphate dehydrogenase | MK388226 | F: AAGCTCAATGGCATTGCACT<br>R: TGGAAGAAACATCAGTGCAC | 58.74<br>56.26 | 1.89 | 205 | 26.87 | 2.33 | 15.64 |
| ACT | Actin | MK388227 | F: TGAGAAGAGCTATGAGTTGC<br>R: GTAGATCCACCACTAAGCAC | 54.89<br>55.27 | 1.98 | 204 | 22.49 | 1.03 | 8.26 |
| TUA | Alpha-tubulin | MK388228 | F: ATATCATCCTTGACAACTTC<br>R: GATGCCACATTTGAAGCCAG | 50.18<br>57.71 | 1.94 | 210 | 22.80 | 1.00 | 7.92 |
| TUB | beta-tubulin | MK388229 | F: GTTAGCGAGCAGTTTACAGC<br>R: ACCAACAACCTCTTCTTCTT | 57.21<br>54.39 | 2.01 | 195 | 23.22 | 1.05 | 8.10 |
| UBC | Ubiquitin-conjugating enzyme E2 | MK388230 | F: CAAATGACAGTCCCTATGCT<br>R: GGTCAGTAAGCAATGAACAT | 55.12<br>53.58 | 1.99 | 219 | 26.05 | 1.81 | 12.48 |
| EF-1α | Elongation factor 1-alpha | MK388231 | F: GACAACGTTGGCTTCAACGT<br>R: GCAAACTTCACAGCAATGTG | 59.62<br>56.43 | 1.92 | 203 | 23.09 | 1.96 | 15.27 |
| TEF | Translation elongation factor | MK388232 | F: GTACAATGATGAGAACACTG<br>R: TCCATGTCATAGCACTCATC | 51.64<br>54.64 | 1.96 | 201 | 22.28 | 1.41 | 11.38 |
| SAND | Sadenosyl methionine decarboxylase | MK388233 | F: GCATCTAGGACAACCTAGAG<br>R: CCACTATCATGCATAGAAGC | 54.38<br>53.65 | 1.86 | 213 | 26.19 | 1.66 | 11.40 |
| YLS8 | Thioredoxin-like protein YLS8 | MK388234 | F: GCATCTGTTGCTGAGACAAT<br>R: CTCCACAATATCAATGAACT | 56.42<br>50.48 | 1.90 | 216 | 23.47 | 1.53 | 11.71 |
| CDPK | CDPK-related kinase | MK388235 | F: GAAGATGAGCTCATCTACCT<br>R: TAAGTACTGATTGCAGCAGC | 53.66<br>55.85 | 1.93 | 212 | 29.80 | 2.06 | 12.45 |

Tm, Melting Temperature; E, PCR amplification efficiency; Ct, cycle threshold; CV, coefficient of variation; SD, standard deviation

[26] software were used to analyze the stabilities of the candidate reference genes' expression. The mean Ct value must be transformed into relative expression level based on the formula $E^{-\Delta Ct}$ ($-\Delta Ct$ = Ct value of each sample-the minimum Ct value) [27]. The $E^{-\Delta Ct}$ value can be analyzed by GeNorm and NormFinder to obtain the candidate reference gene's expression stability. GeNorm can determine the stability of each candidate reference gene through calculating the M value according to its expression. When the M value is less than 1.5, the gene can be used as an appropriate reference gene. Moreover, the lower the M value, the more stable the expression of the candidate reference gene. NormFinder can rank the stability of candidate reference genes expression. Moreover, it can identify the best reference gene among all candidate reference genes. The BestKeeper [28] program analyzed the pairing correlation under a given set of experimental conditions using the Ct values of the reference genes in each group. According to the standard deviation (SD) and coefficient of correlation of the candidate genes R value, the expression stability of each reference gene was evaluated in the BestKeeper program. A reference gene with a lower SD value and an R value closer to 1 would have stable expression. Correlation analysis ($R^2$) using SPSS (19.0) between M values vs SV values, SV value vs SD values, M values vs SD values for determine its data robustness. A simple T test using SPSS to illustrate the significance of data differences. Differences were considered to be significant if $P < 0.05$. * represented $0.01 < P < 0.05$.

## Validation of reference genes

The identified optimum reference genes and the least stable reference genes in salt or in different tissues were used to normalize the relative expression levels of *BpGRAS1* (GenBank

number:MN117546), *BpGRAS16* (GenBank number:MN117547) and *BpGRAS19* (GenBank number:MN117548). *ACT* and *TEF* were used as the most stable reference genes in the control and 200 mM NaCl-treated samples, while *GAPDH* was used as the least stable reference gene. *EF-1α*, *TUB*, and their combinations were used as the most stable reference genes in different birch tissues, while *GAPDH* was used as the least stable reference gene. The same qRT-PCR reaction systems and conditions were used as stated above. Three parallel total RNA samples were applied to analyze the relative expression. Data processing as descried in $2^{-\Delta\Delta Ct}$ [29].

## Results

### Specificity and amplification efficiency of qRT-PCR primers

Agarose gel electrophoresis showed that single PCR products could be amplified from the 11 candidate reference genes (S1 Fig), indicating that the primers designed by Primer 5.0 software had strong specificity and fulfilled the requirements of qRT-PCR primers. The melting curves for all 11 candidate reference genes had a distinct single peak, and the three replications showed perfect repeatability (S2 Fig). The amplification efficiency of these reference genes ranged from 1.86 to 2.01 (Table 1), which met the requirements of the qRT-PCR experiment. Therefore, these subjects reference genes (*ACT*, *TUA*, *TUB*, *TEF*, *18S rRNA*, *EF1α*, *GAPDH*, *UBC*, *YLS8*, *SAND*, and *CDPK*) could serve as the research subjects.

### Analysis of the expression level of the 11 candidate reference genes

Ct value is one of the criteria used to measure gene expression level. A gene with a higher Ct value would have a lower gene expression level [29]. Ct values of the 11 candidate reference genes were unevenly distributed in the different treatments and tissues of birch (Fig 1). However, the mean Ct value can illustrate the abundance of the transcripts from the 11 candidate reference genes (Table 1), which can be used to analyze the stability of reference genes in different experimental samples.

The mean Ct values of all the samples varied from 17.39 to 29.80. Among the 11 candidate reference genes in all samples, 18S *rRNA* had the highest expression level, with an average Ct ± SD of 17.39 ± 1.36. *ACT*, *TUA*, *TUB*, *TEF*, *EF-1α*, and *YLS8* had relatively high expression level, with average Ct ± SD values of 22.49 ± 1.03, 22.80 ± 1.00, 23.23 ± 1.05, 22.28 ± 1.41, 23.09 ±1.96, and 23.47 ± 1.53, respectively. *CDPK* had the highest average Ct value, indicating that it had the lowest expression level (Fig 1).

The lower the coefficient of variation (CV) value, the higher the stability of the candidate reference gene [30]. Among the 11 candidate reference genes in all samples, the CV value of *TUA* was the lowest, followed by *TUB* and *ACT*. *GAPDH* had a high CV value, which indicated that its expression was least stable (Table 1). Thus, the results indicated that the expression levels of 11 candidate reference genes were different among the different experimental conditions.

### GeNorm analysis

The geNorm software can identify reference genes with better stabilities by calculating the average expression stability index (M value). The software will rank candidate reference genes expression stabilities according to their M values. The geNorm algorithm reveals the appropriate number of reference genes required for normalization to calculate the pairwise variation between the normalization factors NFn and NFn+1 of the two sequences (Vn/n+1). When Vn/n+1 is less than 0.15, n is the number of genes used in the normalization [31–32]. The lower the M value, the more stable the reference gene. Conversely, a higher M value indicates

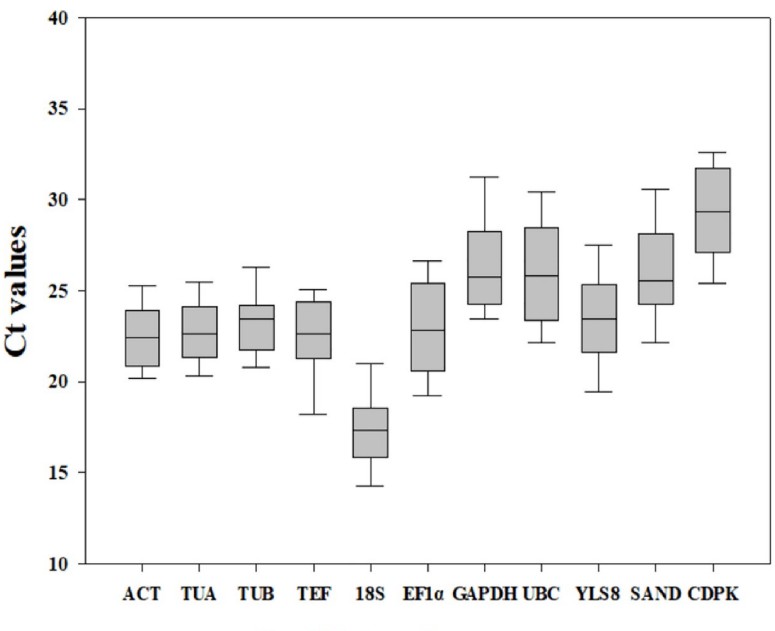

**Fig 1. Cycle threshold (Ct) value distribution of eleven candidate reference genes in different tissues and abiotic stress conditions in birch.** Boxes contain the 5th and 95th quartiles of the Ct values of the different genes tested, median (central horizontal line) and minimal/maximal value (vertical bar).

worse stability. Genes with an M value less than 1.5 could be used as reference genes [33]. To obtain optimum reference genes for the different experimental conditions, total samples were divided into four groups including different tissues, normal, salt, and osmosis, as described in the section describing plant material and treatments. The results are shown in Fig 2. Ten of the eleven candidate reference genes in this study could be considered as appropriate reference genes with the M value less than 1.5, except for *GAPDH* in salt, different tissues, and total samples.

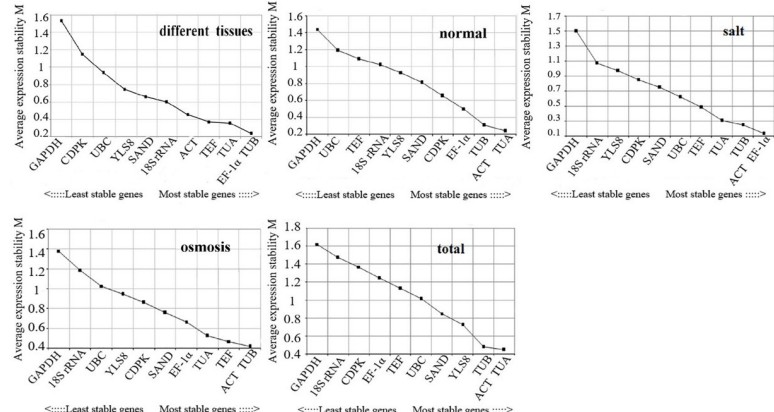

**Fig 2. Expression stability and ranking of eleven candidate reference genes as analyzed by geNorm.** The black spots represent the average expression stability (M). The highest M value indicates the most unstable gene, while the lowest represents the most stable. The order from left to right indicates the stability ranking of the eleven candidate reference genes. The most variable is on the left while the most stable is on the right.

In the different tissues of birch, *TUB* and *EF-1α* were the most stable reference genes. In normal conditions, *ACT* and *TUA* were the most stable. Under salt stress treatment, *ACT* and *EF-1α* were optimum reference genes. Under osmotic stress treatment, *ACT* and *TUB* were the most ideal reference genes. In total samples, *ACT* and *TUA* were the most ideal choices (Fig 2). In conclusion, the most appropriate reference genes were different in the different experimental conditions. However, *ACT* could be the best reference gene in all experimental conditions except in different tissues; *GAPDH* was the least stable reference gene in all the groups in this study.

## NormFinder analysis

NormFinder software can select the most appropriate reference gene according to candidate reference genes expression steady values [34]. A reference gene with the least steady value (SV value) has the most stable expression. NormFinder can not only compare the expression differences of candidate reference genes, but also compares the variation between samples. The results of gene stability ranking of candidate reference genes were generated and are shown in Table 2. *TUB* was the most stable reference gene in total group and in the normal conditions group. *TEF* was optimum reference gene under salt and osmotic stress. *GAPDH* was the least stable reference gene in all experimental groups. *18S rRNA* was unstable in most groups, while it was optimum in the different tissues group. The above results are basically consistent with the geNorm analysis.

## Bestkeeper analysis

Bestkeeper software is commonly used to choose appropriate reference genes; however, the number of candidate reference genes cannot exceed 10. Thus, we chose the 10 best candidate reference genes according to the results of geNorm and NormFinder. In BestKeeper, SD and R values are used to select ideal reference genes, however, the mean standard deviation (mSD) can be used to assess whether the expression of a gene is stable [35]. A gene with an SD value less than 1.0 can be considered as ideal candidate reference genes. At the same time, the closer the R value is to 1, the better the stability of the reference gene. The results of ranking the candidate reference genes according to their SD values are shown in Table 3.

In the different tissues group, it was obvious that *TEF* was the best reference gene, although its R value was far from satisfactory. In normal conditions, *CDPK* had the lowest SD value,

**Table 2. Expression stability values (SV value) of candidate reference genes under different treatments of birch, as calculated using Normfinder.**

| Rank | Different tissues | | Normal conditions | | Salt stress | | Osmotic stress | | Total | |
|---|---|---|---|---|---|---|---|---|---|---|
| | Gene | SV | Gene | SV | Gene | SV | Gene | SV | Gene | SV |
| 1 | 18S rRNA | 0.038 | TUB | 0.356 | TEF | 0.272 | TEF | 0.225 | TUB | 0.302 |
| 2 | EF1α | 0.083 | EF1α | 0.416 | ACT | 0.276 | TUB | 0.289 | ACT | 0.340 |
| 3 | SAND | 0.097 | SAND | 0.418 | TUB | 0.286 | EF1α | 0.387 | TUA | 0.341 |
| 4 | ACT | 0.099 | ACT | 0.503 | EF1α | 0.316 | ACT | 0.428 | YLS8 | 0.562 |
| 5 | TUB | 0.211 | YLS8 | 0.526 | TUA | 0.430 | YLS8 | 0.526 | UBC | 0.741 |
| 6 | YLS8 | 0.458 | TUA | 0.557 | UBC | 0.473 | SAND | 0.547 | SAND | 0.748 |
| 7 | TEF | 0.483 | 18S rRNA | 0.694 | SAND | 0.501 | TUA | 0.624 | TEF | 0.806 |
| 8 | TUA | 0.532 | CDPK | 0.695 | 18S rRNA | 0.777 | CDPK | 0.732 | EF1α | 0.948 |
| 9 | UBC | 1.117 | TEF | 0.745 | YLS8 | 0.889 | UBC | 0.799 | CDPK | 1.091 |
| 10 | CDPK | 1.458 | UBC | 0.968 | CDPK | 0.907 | 18S rRNA | 1.137 | 18S rRNA | 1.097 |
| 11 | GAPDH | 2.226 | GAPDH | 1.685 | GAPDH | 2.328 | GAPDH | 1.438 | GAPDH | 1.399 |

**Table 3. Expression stability values (standard deviation (SD) value and R value) of candidate reference genes under different treatments of birch, as determined by BestKeeper.**

| Rank | Different tissues | | | Normal conditions | | | Salt stress | | | Osmotic stress | | | Total | | |
|---|---|---|---|---|---|---|---|---|---|---|---|---|---|---|---|
| | Gene | SD | R value | Gene | SD | R value | Gene | SD | R value | Gene | SD | R value | Gene | SD | R value |
| 1 | TEF | 0.070 | 0.704 | CDPK | 0.490 | 0.776 | TEF | 0.310 | 0.572 | TUB | 0.310 | 0.955 | ACT | 0.550 | 0.888 |
| 2 | TUA | 0.090 | 0.414 | ACT | 0.630 | 0.912 | TUA | 0.350 | 0.753 | ACT | 0.350 | 0.773 | TUA | 0.550 | 0.891 |
| 3 | TUB | 0.230 | 0.895 | 18S rRNA | 0.650 | 0.694 | ACT | 0.420 | 0.952 | TUA | 0.440 | 0.697 | TUB | 0.550 | 0.881 |
| 4 | EF1α | 0.250 | 0.997 | TUA | 0.710 | 0.909 | TUB | 0.420 | 0.869 | TEF | 0.480 | 0.950 | 18S rRNA | 0.690 | 0.356 |
| 5 | ACT | 0.380 | 0.903 | TUB | 0.740 | 0.950 | EF1α | 0.510 | 0.943 | YLS8 | 0.570 | 0.755 | TEF | 0.710 | 0.662 |
| 6 | 18S rRNA | 0.570 | 0.941 | TEF | 0.950 | 0.901 | UBC | 0.510 | 0.750 | EF1α | 0.630 | 0.944 | YLS8 | 0.810 | 0.877 |
| 7 | SAND | 0.630 | 0.993 | EF1α | 0.970 | 0.957 | SAND | 0.630 | 0.873 | SAND | 0.640 | 0.856 | SAND | 0.880 | 0.793 |
| 8 | UBC | 0.710 | 0.149 | YLS8 | 0.980 | 0.914 | 18S rRNA | 0.730 | 0.348 | CDPK | 0.710 | 0.630 | EF1α | 0.930 | 0.776 |
| 9 | YLS8 | 0.780 | 0.997 | SAND | 1.020 | 0.947 | CDPK | 0.760 | 0.879 | UBC | 1.030 | 0.632 | UBC | 0.970 | 0.822 |
| 10 | CDPK | 0.780 | -0.174 | UBC | 1.120 | 0.848 | YLS8 | 0.860 | 0.853 | 18S rRNA | 1.230 | 0.765 | CDPK | 1.090 | 0.711 |

followed by *ACT*, and the R value of *ACT* was closer to 1 than that of *CDPK*. On the whole, *CDPK* was more appropriate to serve as a candidate reference gene under the normal experimental conditions used. Under salt stress conditions, SD values of all the candidate reference genes were less than 1.0. Thus, the 10 tested candidate reference genes could be used as reference genes, however, the R values of *ACT* and *EF1α* are closer to 1, so we consider choosing them. *TEF* was the best candidate reference gene, which agreed with the results of NormFinder. *ACT* and *EF1α* had R values that closer to 1. Under osmotic stress conditions, *TUB* was the best choice because of its ideal SD and R values. Ultimately, *ACT*, *TUA*, and *TUB* had similar stabilities according to the results in all samples. These three candidate reference genes were suitable as candidate reference genes, which agreed with the results of geNorm.

## Validation of reference genes

To verify the selected reference genes, the most and least stable reference genes were used to assess the relative expression level of target *BpGRAS* genes after normalization under salt stress and different tissues. The GRAS transcription factor family is one of the largest families of transcription factors and are involved in regulating the expression of several target function genes in plants biotic and abiotic responses, including that to salt. High-salt and drought stress could induce the expression of *SCL7* in *Populus euphratica*, which belongs to the GRAS transcription factor family [36]. Eight GRAS transcription factor genes were upregulated in different tissue of *Dendrobium catenatum* following exposure to heat and salt stresses [37]. Our previous research showed that transiently overexpressed *BpGRASs* had significantly increased salt tolerance in birch (unpublished).

We used the most stable and least stable candidate reference genes according to our results to normalize the expression level of *BpGRAS* genes. The expression patterns of the *BpGRAS* genes were markedly different when the most and least stable reference genes were used to normalize their expression in salt stress and different tissues (Fig 3).

When the ideal reference genes *ACT* and *TEF* were used for normalization under treatment with 200 mM NaCl, the expression level of the *BpGRAS* genes were significantly enhanced compared with that in the control group at most time points, and their expression patterns were similar. However, we could not draw the same conclusion when the least stable reference gene was used (Figs 3 and 4). When the ideal reference genes *ACT* and *TEF* were used as internal controls for samples treated with 200 mM NaCl, *BpGRAS1* had at the highest transcript level at 48 h after salt treatment. However, when the least stable reference gene *GAPDH* was

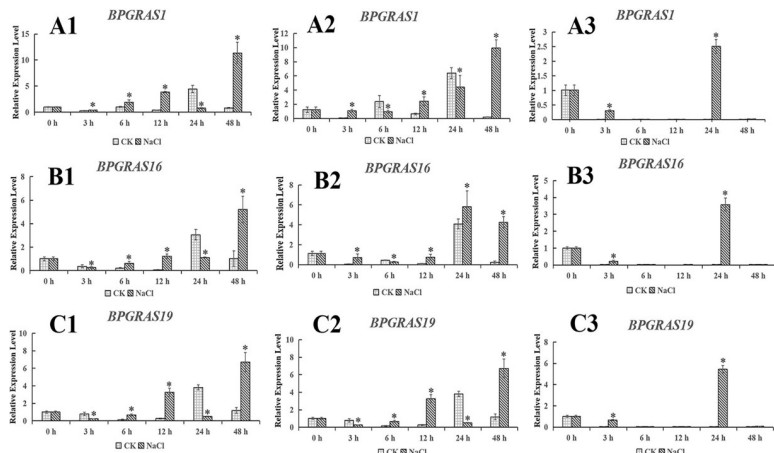

**Fig 3. Relative quantification of targeted *BpGRAS* gene expression levels in birch using validated reference genes.** Samples of leaves collected at 0, 3, 6, 12, 24, and 48 h under salt stress, respectively. A1-C1: Samples treated with 200 mM NaCl using *ACT* as internal the reference gene to normalize the expression of *BpGRAS* genes. A2-C2: Samples treated with 200 mM NaCl using *TEF* as the internal reference gene to normalize the expression of *BpGRAS* genes. A3-C3: Samples treated with 200 mM NaCl using *GAPDH* as the internal reference gene to normalize the expression of *BpGRAS* gene. * represented $0.01 < P < 0.05$.

used to normalize the expression level of *BpGRAS1*, *BpGRAS1* showed it was hardly expressed at 48 h. The results are the same as follows, when the ideal reference genes *ACT* and *TEF* were used for normalization after treatment of 200 mM NaCl, *BpGRAS16* and *BpGRAS19* showed higher transcript levels at 48 h after salt treatment; however, it was hardly expressed at 48 h when the least stable reference gene, *GAPDH*, was used for normalization. Simultaneously, T-test showed that the expression level of NaCl treatment group was significantly higher than that of the control.

The most stable reference genes for use in different tissues were *EF1α* and *TUB* and the least stable reference gene was *GAPDH*. These three genes were used to normalize the expressions of the *BpGRAS* genes. The combination of *EF1α* and *TUB* was also used for normalization. The results are shown in Fig 4. When the appropriate reference genes or their combination were used, the expression of *BpGRAS1* in root was the highest, followed by that in the stem, with the lowest expression in the leaf. The expression of *BpGRAS16* was highest in the stem, followed by leaf and then the root. The expression of *BpGRAS19* in the stem and root were the essentially same. However, when the least stable reference gene was used for normalization, the *BpGRAS* genes were hardly expressed in the stem and root. Obviously, the expression patterns of the *BpGRAS* genes were similar after using the appropriate reference genes and their combinations for normalization. However, the expression patterns of *BpGRAS* genes showed large differences when the least stable reference gene was used.

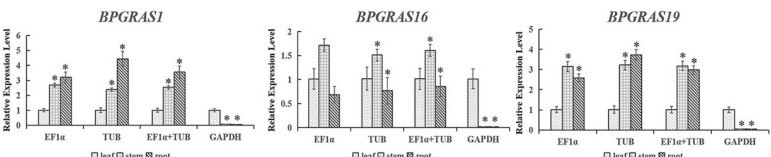

**Fig 4. Expression levels in birch using the validated reference genes.** Samples of leaves, stems, and roots in normal conditions assessed using qRT-PCR with different internal reference gene to normalize the expression of *BpGRAS* genes. * represented $0.01 < P < 0.05$.

In summary, using different reference genes for normalization could produce different experimental results. Therefore, selecting the ideal reference genes according to different experimental conditions and different tissues is very importance.

## Discussion

Determining a gene's expression level is an important step to understand the function of the gene product in biological processes during environmental stress, and in developmental and cellular processes [38]. To ensure the accuracy of relative RT-PCR, specific PCR conditions and an appropriate internal controls must be determined. Thus, the selection of internal reference genes that show stable expression in all tissues and under the experimental conditions being investigated plays a key role in determining stress-related gene expression under different abiotic stress conditions. Identifying suitable internal reference genes in birch will encourage gene expression studies in this tree species.

There are many reports concerning reference gene selection in plant, including in *Salicornia europaea* [12], cucumber [15]. However, until now, there have been no systematic reports about reference genes selection in birch. Therefore, the present study was designed to identify and validate suitable reference genes for gene expression analysis in birch.

In this study, 11 candidate reference genes that have been frequently used in previous studies were evaluated, and three data analysis tools (geNorm, NormFinder, BestKeeper), which are specialized for reference gene selection, were used. The expression levels of the 11 candidate reference genes were significantly different among the different samples (Fig 1). The three analysis tools have different algorithms; therefore, the ranking of these 11 candidate reference genes according to their expression stability varied. The differences in the statistical theories used by the three pieces of software would explain the different results. Using correlation analysis ($R^2$) 11 candidate genes between M values vs SV values (ranged from 0.0022 to 0.8169), SV values vs SD values (ranged from 0.0048 to 0.8232), M values vs SD values (ranged from 0.0068 to 0.9159), which indicated the data could not establish the expression stability. We should select the reference genes with better expression stability as identified by all three tools. Therefore, the rankings of candidate reference genes calculated by these three algorithms are listed (S1 Table). In general, *EF1α* was the best reference gene in different tissues. *ACT* and *TEF* are the best for salt-stressed sample and *TUB* alone is the best for osmotic-stressed sample. Meanwhile, *ACT* was also the most appropriate reference gene in normal conditions and in all the samples. Previously *GAPDH* was identified as the optimum reference gene in salt-treated leaves, Cd-treated roots, cold-treated leaves and roots, and PEG-treated leaf samples in *Carex rigescens* [20]. However, *GAPDH* was the least stable reference gene in all the experimental conditions in the present study, despite being one of the most commonly used reference genes in previous studies.

Suitable reference genes have been confirmed in many plant species in different tissues and under abiotic stress. In *Salicornia europaea*, *ACT* (Actin) and *GAPDH* were the optimum combination of internal reference genes to study gene expression under drought stress [12]. In rice, *UBQ5* and *eEF1α* was most stable as reference genes in different tissues [39]. In tomato, the expression stabilities of *GAPDH* and phosphoglycerate kinase *(PGK)* were ranked as the top during light stress but were poorly ranked during N and cold stress [40]. In cucumber, *TUA* was considered as an appropriate reference gene in different tissues. *EF1α* was identified as a suitable reference gene for abiotic stress treatment [15]. Thus, appropriate reference genes differ among different plants and according to the experimental conditions. In conclusion, different reference genes should be selected according to different experimental conditions to obtain accurate results.

## Conclusion

In this study, *ACT* was identified as the optimum reference gene in all experimental groups, except in the different tissues group. *GAPDH* was the least stable candidate reference gene in all experimental conditions. This study provided appropriate reference genes for expression studies in birch, which will be beneficial for more accurate relative quantification of mRNA expression in birch for different tissues, in normal conditions, and under salt and osmotic stress conditions.

## Supporting information

**S1 Fig. The results of 1.5% agarose gel electrophoresis after 30 cycles PCR.**
(TIF)

**S2 Fig. Dissociation curves of the qRT-PCR amplicons.**
(TIF)

**S1 Table. Comprehensive rankings of the stability of the reference genes by three algorithms.** GN: Ranking of candidate reference genes calculated by geNorm. NF: Ranking of candidate reference genes calculated by NormFinder. BK: Ranking of candidate reference genes calculated by Bestkeeper
(DOCX)

## Author Contributions

**Conceptualization:** Ziyi Li, Yucheng Wang, Xiaoyu Ji.

**Data curation:** Ziyi Li.

**Formal analysis:** Ziyi Li, Huijun Lu, Zihang He.

**Funding acquisition:** Xiaoyu Ji.

**Investigation:** Huijun Lu.

**Methodology:** Xiaoyu Ji.

**Project administration:** Chao Wang, Yucheng Wang.

**Software:** Ziyi Li, Huijun Lu, Zihang He.

**Supervision:** Yucheng Wang.

**Validation:** Zihang He, Chao Wang.

**Visualization:** Huijun Lu, Zihang He, Chao Wang.

**Writing – original draft:** Ziyi Li, Huijun Lu.

**Writing – review & editing:** Ziyi Li.

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
