## [Decision Letter · Decision Letter 0]

7 Oct 2019

PONE-D-19-25334

Selection of appropriate reference genes for quantitative real-time reverse transcription PCR in Betula platyphylla under salt and osmotic stress conditions

PLOS ONE

Dear Dr Xiaoyu Ji,

Thank you for submitting your manuscript to PLOS ONE. After careful consideration, we feel that it has merit but does not fully meet PLOS ONE’s publication criteria as it currently stands. Therefore, we invite you to submit a revised version of the manuscript that addresses the points raised during the review process.

We would appreciate receiving your revised manuscript by 19th October 2019. To enhance the reproducibility of your results, we recommend that if applicable you deposit your laboratory protocols in protocols.io, where a protocol can be assigned its own identifier (DOI) such that it can be cited independently in the future. For instructions see: http://journals.plos.org/plosone/s/submission-guidelines#loc-laboratory-protocols

We look forward to receiving your revised manuscript.

Kind regards,

Mayank Gururani

Academic Editor

PLOS ONE

Journal Requirements:

2.  PLOS ONE now requires that authors provide the original uncropped and unadjusted images underlying all blot or gel results reported in a submission’s figures or Supporting Information files. This policy and the journal’s other requirements for blot/gel reporting and figure preparation are described in detail at https://journals.plos.org/plosone/s/figures#loc-blot-and-gel-reporting-requirements and https://journals.plos.org/plosone/s/figures#loc-preparing-figures-from-image-files. When you submit your revised manuscript, please ensure that your figures adhere fully to these guidelines and provide the original underlying images for all blot or gel data reported in your submission. See the following link for instructions on providing the original image data: https://journals.plos.org/plosone/s/figures#loc-original-images-for-blots-and-gels

-http://hsianglab.000webhostapp.com/pubs/pdf/02rtpcr_pmbr.pdf

-https://www.ncbi.nlm.nih.gov/pmc/articles/PMC4217110/

-https://www.frontiersin.org/articles/10.3389/fpls.2017.01659/full

In your revision ensure you cite all your sources (including your own works), and quote or rephrase any duplicated text outside the methods section. Further consideration is dependent on these concerns being addressed.

Additional Editor Comments (if provided):

Reviewers' comments:

Reviewer's Responses to Questions

**Comments to the Author**

1. Is the manuscript technically sound, and do the data support the conclusions?

Reviewer #1: Partly

Reviewer #2: Yes

2. Has the statistical analysis been performed appropriately and rigorously? 

Reviewer #1: No

Reviewer #2: Yes

3. Have the authors made all data underlying the findings in their manuscript fully available?

Reviewer #1: Yes

Reviewer #2: Yes

4. Is the manuscript presented in an intelligible fashion and written in standard English?

Reviewer #1: Yes

Reviewer #2: Yes

5. Review Comments to the Author

Reviewer #1: An impressive body of work is presented in this manuscript to select appropriate reference genes in birch. Overall, the manuscript is reasonably well written. However, there are some points that the authors need to consider.

Critical points:

1. Multiple parameters need to consider before choosing an ideal reference gene for qPCR analysis. Among them, one of the very important criteria is qPCR efficiency, and it should be close to 2 (within 80–100%). The authors used E-ΔCt formula for data analysis. I assume ‘E’ stands for qPCR efficiency. Therefore, it is necessary to measure the qPCR efficiency of each gene and mention in the manuscript. Apart from that, all other criteria are satisfactory: single pick dissociation curve, single band in agarose gel, and amplicon size.

2. Authors tried to summarize the analysis of three statistical algorithms (GeNorm, NormFinder, and BestKeeper) by calculating the ‘sum of rankings’ in table 4. Before that, it needs to establish the robustness of data by correlation analysis (R2). Example: correlation analysis between M Value vs Stability Value. After that, authors can perform a sum of rankings analysis using the best-correlated algorithms or all algorithms.

3. Statistical analysis is missing in figure 3 and 4. I would suggest performing a simple T-test.

4. Sampling and calculation procedure is not clear. I assume authors harvested root, stem and leaf tissues from normal and stressed samples after the given time points. How many samples were used per experiments? Are they biological or technical replication? Also, maintain uniformity in the text; either use “Normal” or “Control”. In addition, authors should clarify how they calculate average for each groups (i.e., tissue, normal, salt and osmotic).

Minor points:

Line 65: All are not woody plants. Example. Musa paradisiaca

Line 100: Mention cycle no in the Fig.A1 legend.

Line 113: Range of the mean Ct values (17.34 to 28.27) is not matching with Table 1 (17.39 to 29.80).

Line 138: To follow the result easily, mention the “four groups” here as well. Also, describe “total” sample.

Line 142-145: Some data points are missing in Figure 2. Ex: EF-1α in different tissues and ACT in salt stress.

Line 145: Use uniform terminology in the manuscript. Either “all samples” or “total samples”.

Line 151: Authors should explain the calculation procedures in details in Figure 2 legend. How they calculated “the average expression stability (M)” values for normal and stressed samples? I assume the authors averaged the data from different time points as well as different tissues. Also, they should mention the replication no. I would recommend this suggestion for other figures and tables as well.

Line 132 & 155: Numbers are same for both sections.

Line 172: Are not “M values” stands for geNorm analysis? If so, correct Table 2 as it is showing values from NormFinder analysis.

Line 183: I think authors should explain why they choose SD value over the R-value for the ranking.

Line 186-187: “On the whole…. experimental conditions used”; Do you mean normal condition?

Line 188 -189: “Under salt stress conditions…. used as reference genes”; In this situation authors can consider the R-value to choose best one.

Line 192: “Ultimately, ACT…. to the results”; Mention the sample name.

Line 202: The authors validated well the best reference genes through expression analysis of GRAS TFs in salt-stressed samples. To do that, the authors cited multiple references that showed stressed-mediated expression of GRAS homologs. However, they did not cite any reference for tissue-specificity. I would recommend adding some references to show that the GRAS TFs can be expressed tissue-specific manner.

Line 255: I think it will be “A2-C2” instead of “B2-C2”.

Line 293: Authors mentioned about cloning. However, no such section available in the materials and method.

Line 303. Why not TUB for osmotic stress. It showed lowest sum of rankings.

Line 342-349: Mention RNA amount that were used for cDNA synthesis.

Line 351-356: Mention Tm of the primers.

Line 367-368: Mention cut-off/ threshold for the Ct values (if manually adjusted).

Line 370: How many biological and/or technical replications were used for mean Ct value calculation?

Reviewer #2: The authors have clearly established standard reference gene(s) that can be used for qRT-PCR evaluation in the widely used timber tree, Birch (Betula platyphylla). The design of experiments and the candidate genes chosen for the study follows the standard studies that has been performed in several plant and animal species. The results are clearly expressed.

General comments:

1. The description of results and discussion are jittered and need to be rephrased to get a seamless flow.

2. Excessive references (65) that doesn’t add more value to the paper may be restricted to max.40. More of recent and references of woody species should be included eg.:

a. Wang JJ, Han S, Yin W, Xia X, Liu C. Comparison of Reliable Reference Genes Following Different Hormone Treatments by Various Algorithms for qRT-PCR Analysis of Metasequoia. International Journal of Molecular Sciences. 2018. 20(1):34

b. Wei Y, Liu Q, Dong H, et al. Selection of Reference Genes for Real-Time Quantitative PCR in Pinus massoniana Post Nematode Inoculation. PLoS One. 2016;11(1):e0147224.

c. Wang J, Abbas M, Wen Y, Niu D, Wang L, Sun Y, et al. (2018) Selection and validation of reference genes for quantitative gene expression analyses in black locust (Robinia pseudoacacia L.) using real-time quantitative PCR. PLoS ONE 13(3): e0193076. https://doi.org/10.1371/journal.pone.0193076

d. Chen X, Mao Y, Huang S, et al. Selection of Suitable Reference Genes for Quantitative Real-time PCR in Sapium sebiferum. Front Plant Sci. 2017;8:637.

3. The results, few sentences verbatim is repeated in the abstract, results and discussion – which might be rephrased.

However a few points, listed below, when rectified in the paper would be perfect.

Major :

Line 65 – Ref 29 is either misplaced/ missing

Line 66 – Triticum is not a woody plant – and including this example doesn’t add any new information.

In the - ‘Validation of reference genes’ results section paragraphs starting from line 220 to 235 is not clear and the reader gets lost. Consider rephrasing the paras.

Line 287 – Too many references may be avoided, to keep it concise. The given literature has been previously cited in the introduction section also. Eg.: Line 288 – reference 52 – the original paper has No PCR studies reported in this paper and hence irrelevant.

Line 317 – wrong reference cited (44) instead it should be 62.

Line 321 – ref. Wan et al., 2010 – not in reference.

Line 323 – wrong reference – referring to animal sciences paper not Magnolia denudate

Line 334 – the source of birch seeds should be mentioned.

Line 337 – the quantity (in ml or L) used each time may be mentioned. Whether tap water or distilled water was used?

Line 424 – ref.10 – non relevant animal studies paper may be omitted.

Line 475 – ref.29 is NOT a ‘validation’ paper

Line 536 – ref.52 - no RT-PCR studies reported in the paper, hence irrelevant.

Line 567 – ref.63 – irrelevant reference

Minor:

Line 39 – maybe modified as “food spoilage detection” to be specific.

Line 40 – just TWO important references would suffice to support.

Line 74 and 78 – not clear – consider rephrasing.

Line 80 – a couple of recent literature would suffice.

Line 86 – The sentence is repeated in the abstract verbatim, which may be avoided.

Line 92 – “reached” may be replaced with “fulfilled”

Line 94 – “repeats” may be replaced with “replications”

Line 96 – “subjects” may be replaced with “candidates”

Superfluous ‘the’ may be omitted – eg. Line 106 and few other paragraphs.

Line 118 – The result is repeated elsewhere and also in the same para line 114.

Line 340 – ‘cryotheraphy’ may be replaced with ‘after freezing in liquid nitrogen’

Line 341 – ‘experiments’ may be replaced with ‘replications’

Line 344 – ‘detected’ may be replaced with ‘checked’

Line 350 – ‘synthetic’ may be replaced with ‘synthesized’

Line 353 – too many references, repeated.

Line 365 – ‘experiments’ may be replaced with ‘replications’

Line 393 – the non-standard phrase ‘depended on the way of’ may be replaced with ‘as described in’

6. PLOS authors have the option to publish the peer review history of their article (what does this mean?). If published, this will include your full peer review and any attached files.

Reviewer #1: No

Reviewer #2: Yes: Rajagopal Bala

---

## [Author Response · Author response to Decision Letter 0]

29 Oct 2019

Dear reviewers,

We are very grateful to your and the editors’ valuable suggestions. Based on these suggestions, we have made careful modification on the original manuscript. We also responded point by point to the reviewer’s comments as listed below. Hope these will make it more acceptable for publication. If there are further issues to be clarified, please contact us without hesitation. Thank you.

With best wishes,

Xiaoyu Ji

Reviewer 1

Critical points:

1. Multiple parameters need to consider before choosing an ideal reference gene for qPCR analysis. Among them, one of the very important criteria is qPCR efficiency, and it should be close to 2 (within 80–100%). The authors used E-ΔCt formula for data analysis. I assume ‘E’ stands for qPCR efficiency. Therefore, it is necessary to measure the qPCR efficiency of each gene and mention in the manuscript. Apart from that, all other criteria are satisfactory: single pick dissociation curve, single band in agarose gel, and amplicon size. 

Response: We have added PCR efficiency (E) in Table 1.(see line: 152-154, 233-234.)

2. Authors tried to summarize the analysis of three statistical algorithms (GeNorm, NormFinder, and BestKeeper) by calculating the ‘sum of rankings’ in table 4. Before that, it needs to establish the robustness of data by correlation analysis (R2). Example: correlation analysis between M Value vs Stability Value. After that, authors can perform a sum of rankings analysis using the best-correlated algorithms or all algorithms.

Response: We have established the robustness of data by correlation analysis (R2) between M Value vs Stability Value, M Value vs Stability Value. (see line: 176-177, 400-402.)

3. Statistical analysis is missing in figure 3 and 4. I would suggest performing a simple T-test.

Response: We have added it according to the suggestion. T-test was performed to analyze its significanced, and the results were shown in Figure 3 and Figure 4. (see line: 357-371.)

4. Sampling and calculation procedure is not clear. I assume authors harvested root, stem and leaf tissues from normal and stressed samples after the given time points. How many samples were used per experiments? Are they biological or technical replication? Also, maintain uniformity in the text; either use “Normal” or “Control”. In addition, authors should clarify how they calculate average for each groups (i.e., tissue, normal, salt and osmotic).

Response: We have explained it according to the suggestion. Biologically repeated sampling from 6 birch seedlings in three independent experiments. Three independent replications were performed for technical repetition in qRT-PCR. The mean value is arithmetic mean.(see line: 120, 154.) 

Minor points:

1. Line 65: All are not woody plants. Example. Musa paradisiaca. 

Response: We have deleted the relevant reference according to the suggestion. (see line: 87.) 

2. Line 100: Mention cycle no in the Fig.A1 legend.

Response: We have added mentioned cycle in the Figure S1 according to the suggestion. (see line: 202.)

3. Line 113: Range of the mean Ct values (17.34 to 28.27) is not matching with Table 1 (17.39 to 29.80).

Response: We have corrected it according to the suggestion. The range of the mean Ct values is 17.39~29.80. (see line: 214.)

4. Line 138: To follow the result easily, mention the “four groups” here as well. Also, describe “total” sample.

Response: Four groups means ‘different tissues, normal, salt, and osmosis’. Total number of samples called ‘total sample’, and we have corrected it according to the suggestion. (see line: 245-246.)

5. Line 142-145: Some data points are missing in Figure 2. Ex: EF-1α in different tissues and ACT in salt stress.

Response: Missing data points (EF-1α in different tissues, ACT in salt stress, ACT in normal condition) have been completed. We have reconstructed the Figure 2 according to the suggestion. 

6. Line 145: Use uniform terminology in the manuscript. Either “all samples” or “total samples”.

Response: We have corrected it according to the suggestion. Use uniform terminology ‘total sample’. (see line :253.)

7. Line 151: Authors should explain the calculation procedures in details in Figure 2 legend. How they calculated “the average expression stability (M)” values for normal and stressed samples? I assume the authors averaged the data from different time points as well as different tissues. Also, they should mention the replication no. I would recommend this suggestion for other figures and tables as well.

Response: We have added it according to the suggestion. Detailed calculation process has been demonstrated. (see line :238-242.)

8. Line 132 & 155: Numbers are same for both sections.

Response: We have corrected number according to the suggestion.(see line: 206, 235.)

9. Line 172: Are not “M values” stands for geNorm analysis? If so, correct Table 2 as it is showing values from NormFinder analysis.

Response: The Normfinder software processing results are SV values. We have corrected it according to the suggestion. (see line :274.)

10. Line 183: I think authors should explain why they choose SD value over the R-value for the ranking.

Response: More references were ranked with SD values, which was more convincing, however R values can also be used as ranking criteria. We have explained it in the revised manuscript. (see line :288-290.)

11. Line 186-187: “On the whole…. experimental conditions used”; Do you mean normal condition?

Response: Yes, it was normal condition. We have corrected it according to the suggestion. (see line :298.)

12. Line 188-189: “Under salt stress conditions…. used as reference genes”; In this situation authors can consider the R-value to choose best one.

Response: We have taken this method according to the suggestion. Compared R values in this case and find that the R values of ACT and EF1α were closer to 1, so we choosed them. (see line :300-301.)

13. Line 192: “Ultimately, ACT…. to the results”; Mention the sample name.

Response: We have corrected it according to the suggestion. (see line :304.)

14. Line 202: The authors validated well the best reference genes through expression analysis of GRAS TFs in salt-stressed samples. To do that, the authors cited multiple references that showed stressed-mediated expression of GRAS homologs. However, they did not cite any reference for tissue-specificity. I would recommend adding some references to show that the GRAS TFs can be expressed tissue-specific manner.

Response: We have cited relevant reference to illustrate the tissue specificity of GRAS TFs expression according to the suggestion. (see line :318-320.)

15. Line 255: I think it will be “A2-C2” instead of “B2-C2”.

Response: We have corrected it according to the suggestion. (see line :362.)

16. Line 293: Authors mentioned about cloning. However, no such section available in the materials and method.

Response: In the previous study, we have obtained the sequence of these 11 genes. We have mentioned it according to the suggestion. (see line :394.)

17. Line 303: Why not TUB for osmotic stress. It showed lowest sum of rankings.

Response: TUB ranks better than TEF. We have corrected it according to the suggestion. (see line :45, 405.)

18. Line 342-349: Mention RNA amount that were used for cDNA synthesis.

Response: We have added it in the material method according to the suggestion. (see line :130-131.)

19. Line 351-356: Mention Tm of the primers.

Response: We have added it in Table 1 according to the suggestion. (see line :233.)

20. Line 367-368: Mention cut-off/ threshold for the Ct values (if manually adjusted).

Response: The cycle threshold (Ct) values is the machine setting. We have explained it in the revised manuscript. (see line :159-160.)

21. Line 370: How many biological and/or technical replications were used for mean Ct value calculation?

Response: The mean Ct value was calculated using three biological replications, we have added it according to the suggestion. (see line :160.)

Reviewer 2

General comments:

1. The description of results and discussion are jittered and need to be rephrased to get a seamless flow.

Response: We have reworded inappropriate and redundant parts according to the suggestion.

2. Excessive references (65) that doesn’t add more value to the paper may be restricted to max.40. More of recent and references of woody species should be included eg.:

a. Wang JJ, Han S, Yin W, Xia X, Liu C. Comparison of Reliable Reference Genes Following Different Hormone Treatments by Various Algorithms for qRT-PCR Analysis of Metasequoia. International Journal of Molecular Sciences. 2018. 20(1):34.

b. Wei Y, Liu Q, Dong H, et al. Selection of Reference Genes for Real-Time Quantitative PCR in Pinus massoniana Post Nematode Inoculation. PLoS One. 2016;11(1):e0147224.

c. Wang J, Abbas M, Wen Y, Niu D, Wang L, Sun Y, et al. (2018) Selection and validation of reference genes for quantitative gene expression analyses in black locust (Robinia pseudoacacia L.) using real-time quantitative PCR. PLoS ONE 13(3): e0193076. https://doi.org/10.1371/journal.pone.0193076

d. Chen X, Mao Y, Huang S, et al. Selection of Suitable Reference Genes for Quantitative Real-time PCR in Sapium sebiferum. Front Plant Sci. 2017;8:637.

Response: We have read these four articles in detail and cited them, while ensuring that the references are not redundant, within 40 articles according to the suggestion.

3. The results, few sentences verbatim is repeated in the abstract, results and discussion – which might be rephrased.

Response: We have corrected them according to the suggestion, eg.: see line: 108. 

Major:

1. Line 65 – Ref 29 is either misplaced/ missing

Response: We have deleted it according to the suggestion. (see line :87.)

2. Line 66 – Triticum is not a woody plant – and including this example doesn’t add any new information.

Response: We have deleted it according to the suggestion. (see line :89.)

3. In the - ‘Validation of reference genes’ results section paragraphs starting from line 220 to 235 is not clear and the reader gets lost. Consider rephrasing the paras.

Response: We have corrected it and deleted corresponding unclear segment according to the suggestion. (see line :334-336.)

4. Line 287 – Too many references may be avoided, to keep it concise. The given literature has been previously cited in the introduction section also. Eg.: Line 288 – reference 52 – the original paper has No PCR studies reported in this paper and hence irrelevant.

Redundant reference has been.

Response: We have deleted it according to the suggestion.(see line :388-389.)

5. Line 317 – wrong reference cited (44) instead it should be 62. 

Response: We have corrected it according to the suggestion. (see line : 416.)

6. Line 321 – ref. Wan et al., 2010 – not in reference.

Response: We have cited this reference according to the suggestion. (see line : 419.)

7. Line 323 – wrong reference – referring to animal sciences paper not Magnolia denudate.

Response: We have deleted it according to the suggestion. (see line : 420.)

8. Line 334 – the source of birch seeds should be mentioned.

Response: We have mentioned birch seeds source in the revised manuscript. (see line : 113-114.)

9. Line 337 – the quantity (in ml or L) used each time may be mentioned. Whether tap water or distilled water was used?

Response: We have mentioned volume of distilled water according to the suggestion. (see line : 117-118.)

10. Line 424 – ref.10 – non relevant animal studies paper may be omitted.

Response: We have deleted it according to the suggestion. 

11. Line 475 – ref.29 is NOT a ‘validation’ paper.

Response: We have deleted it according to the suggestion. 

12. Line 536 – ref.52 - no RT-PCR studies reported in the paper, hence irrelevant.

Response: We have deleted it according to the suggestion. 

13. Line 567 – ref.63 – irrelevant reference.

Response: We have deleted it according to the suggestion. 

Minor:

1. Line 39 – maybe modified as “food spoilage detection” to be specific.

Response: We have corrected it according to the suggestion. (see line : 61.)

2. Line 40 – just TWO important references would suffice to support.

Response: We have deleted redundant references according to the suggestion. (see line : 62.)

3. Line 74 and 78 – not clear – consider rephrasing.

Response: We have rephrased it according to the suggestion.(see line : 94-98.)

4. Line 80 – a couple of recent literature would suffice.

Response: We have deleted redundant references according to the suggestion.(see line : 101.)

5. Line 86 – The sentence is repeated in the abstract verbatim, which may be avoided.

Response: We have rephrased repeated words according to the suggestion. (see line : 108.)

6. Line 92 – “reached” may be replaced with “fulfilled”

Response: We have corrected it according to the suggestion. (see line : 196.)

7. Line 94 – “repeats” may be replaced with “replications”

Response: repeats  replications. We have corrected it according to the suggestion. (see line : 198.)

8. Line 96 – “subjects” may be replaced with “candidates”

Response: We have corrected it according to the suggestion. (see line : 199.)

9. Superfluous ‘the’ may be omitted – eg. Line 106 and few other paragraphs.

Response: We have corrected it according to the suggestion. (see line : 208, 209.)

10. Line 118 – The result is repeated elsewhere and also in the same para line 114.

Response: We have removed repeated results according to the suggestion. (see line : 218-219.)

11. Line 340 – ‘cryotheraphy’ may be replaced with ‘after freezing in liquid nitrogen’

Response: We have corrected it according to the suggestion. (see line : 121.)

12. Line 341 – ‘experiments’ may be replaced with ‘replications’

Response: experiments  replications.We have corrected it according to the suggestion. (see line : 120.)

13. Line 344 – ‘detected’ may be replaced with ‘checked’

Response: We have corrected it according to the suggestion. (see line : 127.)

14. Line 350 – ‘synthetic’ may be replaced with ‘synthesized’

Response: We have corrected it according to the suggestion. (see line : 133.)

15. Line 353 – too many references, repeated.

Response: We have deleted redundant references according to the suggestion. (see line : 138.)

16. Line 365 – ‘experiments’ may be replaced with ‘replications’

Response: experiments  replications. We have corrected it according to the suggestion. (see line : 154.)

17. Line 393 – the non-standard phrase ‘depended on the way of’ may be replaced with ‘as described in’

Response: We have corrected it according to the suggestion. (see line : 190.)

---

## [Decision Letter · Decision Letter 1]

5 Nov 2019

PONE-D-19-25334R1

Selection of appropriate reference genes for quantitative real-time reverse transcription PCR in Betula platyphylla under salt and osmotic stress conditions

PLOS ONE

Dear Dr. Ji

Thank you for submitting your manuscript to PLOS ONE. After careful consideration, we feel that it has merit but does not fully meet PLOS ONE’s publication criteria as it currently stands. Therefore, we invite you to submit a revised version of the manuscript that addresses the points raised during the review process.

We would appreciate receiving your revised manuscript by 15th Nov. 2019. To enhance the reproducibility of your results, we recommend that if applicable you deposit your laboratory protocols in protocols.io, where a protocol can be assigned its own identifier (DOI) such that it can be cited independently in the future. For instructions see: http://journals.plos.org/plosone/s/submission-guidelines#loc-laboratory-protocols

We look forward to receiving your revised manuscript.

Kind regards,

Mayank Gururani

Academic Editor

PLOS ONE

Reviewers' comments:

Reviewer's Responses to Questions

**Comments to the Author**

1. If the authors have adequately addressed your comments raised in a previous round of review and you feel that this manuscript is now acceptable for publication, you may indicate that here to bypass the “Comments to the Author” section, enter your conflict of interest statement in the “Confidential to Editor” section, and submit your "Accept" recommendation.

Reviewer #1: (No Response)

2. Is the manuscript technically sound, and do the data support the conclusions?

Reviewer #1: Yes

3. Has the statistical analysis been performed appropriately and rigorously? 

Reviewer #1: Yes

4. Have the authors made all data underlying the findings in their manuscript fully available?

Reviewer #1: Yes

5. Is the manuscript presented in an intelligible fashion and written in standard English?

Reviewer #1: Yes

6. Review Comments to the Author

Reviewer #1: A greatly improved manuscript from the first version. The text is now mostly a coherent and well-written piece of work. I would be happy to accept this work, with some minor revisions as indicated here:

1. Line 175-176: Author should add the graph for correlation analysis or at least mention the R2 value in the manuscript.

2. Line 337-339: “Simultaneously, T-test showed that the expression of GAPDH was significantly lower than other reference genes” – I think this is an incorrect statement. T-test result is showing the significant difference of GRAS expression between control vs salt-treated sample. Thus, only the second part of this sentence is appropriate – “the expression level of NaCl treatment group was significantly higher than that of the control”.

3. Author should mention the P-value threshold for T-test in Figure 3 and 4 legends.

4. Line 403-404: “ACT and TUB were the optimum reference genes under salt and osmotic stress.” However, according to Table 4, ACT and TEF are the best for salt-stressed sample and TUB alone is the best for osmotic-stressed sample. Author should cross-check it and the Abstract as well (Line 44-45).

5. According to authors (line 117-119), there are three treatment groups (i.e., Control, Salt, and Osmotic), five time points (3, 6, 12, 24, and 48 h), and three tissue samples (i.e., root, stem, and leaf). Which tissue samples were used for figure 3? Similarly, which treatment groups and what time points were used for Figure 4? Author should clarify it in the respective figure legend.

7. PLOS authors have the option to publish the peer review history of their article (what does this mean?). If published, this will include your full peer review and any attached files.

Reviewer #1: Yes: Ritesh Ghosh

---

## [Author Response · Author response to Decision Letter 1]

14 Nov 2019

Reviewer 1

1. Line 175-176: Author should add the graph for correlation analysis or at least mention the R2 value in the manuscript.

Response: We have analyzed the data of 11 candidate genes between M values vs SV values, SV values vs SD values , M values vs SD values, which indicated the data could not establish the expression stability. We have been added the scope of R2 to the discussion section. (see line: 176-177, 406-409) The data of R2 in detail as follows.

Gene R2 (M vs SV) R2 (SD vs SV) R2 (M vs SD)

CDPK 0.5082 0.2565 0.8657

YSL8 0.2269 0.0534 0.0068

SAND 0.7552 0.1032 0.4931

UBC 0.3972 0.1771 0.9159

TEF 0.8121 0.4036 0.8018

18S 0.8169 0.416 0.1001

EF1α 0.8158 0.6027 0.93

ACT 0.068 0.2959 0.0611

TUA 0.0022 0.0048 0.0315

TUB 0.2061 0.8232 0.0996

We have calculated the ranking of candidate reference genes by three algorithms according to the following references:

1.Wang J, Abbas M, Wen Y, Niu D, Wang L, Sun Y, et al. (2018) Selection and validation of reference genes for quantitative gene expression analyses in black locust (Robinia pseudoacacia L.) using real-time quantitative PCR. PLoS ONE 13(3): e0193076.

2.He M, Cui S, Yang X, Mu G, Chen H and Liu L. (2017) Selection of suitable reference genes for abiotic stress-responsive gene expression studies in peanut by real-time quantitative PCR. Electronic Journal of Biotechnology 28: 76–86. 

3.Wei Y, Liu Q, Dong H, Zhou Z, Hao Y, Chen X, et al. (2016) Selection of Reference Genes for Real-Time Quantitative PCR in Pinus massoniana. PLoS One 11(1): e0147224. 

4.Zhu J, Zhang L, Li W, Han S, Yang W and Qi L (2013) Reference Gene Selection for Quantitative Real-time PCR Normalization in Caragana intermedia under Different Abiotic Stress Conditions. PLoS One 8(1): e53196.

2. Line 337-339: “Simultaneously, T-test showed that the expression of GAPDH was significantly lower than other reference genes” – I think this is an incorrect statement. T-test result is showing the significant difference of GRAS expression between control vs salt-treated sample. Thus, only the second part of this sentence is appropriate – “the expression level of NaCl treatment group was significantly higher than that of the control”.

Response: We have deleted this inappropriate sentence according to the suggestion. (see line: 342-344)

3. Author should mention the P-value threshold for T-test in Figure 3 and 4 legends.

Response: We have added P-value threshold for T-test in the Figure 3 and 4 legends according to the suggestion. (see line: 178-179, 368, 373-374)

4. Line 403-404: “ACT and TUB were the optimum reference genes under salt and osmotic stress.” However, according to Table 4, ACT and TEF are the best for salt-stressed sample and TUB alone is the best for osmotic-stressed sample. Author should cross-check it and the Abstract as well (Line 44-45).

Response: Yes, TUB alone is the best for osmotic-stressed sample. We have cross-checked the manuscript and corrected it according to the suggestion. (see line: 44-45, 412-413)

5. According to authors (line 117-119), there are three treatment groups (i.e., Control, Salt, and Osmotic), five time points (3, 6, 12, 24, and 48 h), and three tissue samples (i.e., root, stem, and leaf). Which tissue samples were used for figure 3? Similarly, which treatment groups and what time points were used for Figure 4? Author should clarify it in the respective figure legend.

Response: Leaf tissue samples were used in the Figure 3, and leaves, stems, and roots in normal conditions samples were used in the Figure 4. We have added it in the Figure 3 and Figure 4 legends according to the suggestion. (see line: 363, 372)

---

## [Decision Letter · Decision Letter 2]

18 Nov 2019

Selection of appropriate reference genes for quantitative real-time reverse transcription PCR in Betula platyphylla under salt and osmotic stress conditions

PONE-D-19-25334R2

Dear Dr. Ji,

We are pleased to inform you that your manuscript has been judged scientifically suitable for publication and will be formally accepted for publication once it complies with all outstanding technical requirements.

With kind regards,

Mayank Gururani

Academic Editor

PLOS ONE

Additional Editor Comments (optional):

Reviewers' comments:

Reviewer's Responses to Questions

**Comments to the Author**

1. If the authors have adequately addressed your comments raised in a previous round of review and you feel that this manuscript is now acceptable for publication, you may indicate that here to bypass the “Comments to the Author” section, enter your conflict of interest statement in the “Confidential to Editor” section, and submit your "Accept" recommendation.

Reviewer #1: All comments have been addressed

2. Is the manuscript technically sound, and do the data support the conclusions?

Reviewer #1: Yes

3. Has the statistical analysis been performed appropriately and rigorously? 

Reviewer #1: Yes

4. Have the authors made all data underlying the findings in their manuscript fully available?

Reviewer #1: Yes

5. Is the manuscript presented in an intelligible fashion and written in standard English?

Reviewer #1: Yes

6. Review Comments to the Author

Reviewer #1: (No Response)

7. PLOS authors have the option to publish the peer review history of their article (what does this mean?). If published, this will include your full peer review and any attached files.

Reviewer #1: Yes: Ritesh Ghosh

---

## [Editor Report · Acceptance letter]

22 Nov 2019

PONE-D-19-25334R2 

Selection of appropriate reference genes for quantitative real-time reverse transcription PCR in *Betula platyphylla* under salt and osmotic stress conditions 

Dear Dr. Ji:

I am pleased to inform you that your manuscript has been deemed suitable for publication in PLOS ONE. Congratulations! Your manuscript is now with our production department. 

With kind regards,

on behalf of

Dr. Mayank Gururani 

Academic Editor

PLOS ONE